# Precision-Recall Balanced Topic Modelling

**Seppo Virtanen**
University of Cambridge
sjv35@cam.ac.uk

**Mark Girolami**
University of Cambridge and The Alan Turing Institute
mag92@cam.ac.uk

## Abstract

Topic models are becoming increasingly relevant probabilistic models for dimensionality reduction of text data, inferring topics that capture meaningful themes of frequently co-occurring terms. We formulate topic modelling as an information retrieval task, where the goal is, based on the latent topic representation, to capture relevant term co-occurrence patterns. We evaluate performance for this task rigorously with regard to two types of errors, false negatives and positives, based on the well-known precision-recall trade-off and provide a statistical model that allows the user to balance between the contributions of the different error types. When the user focuses solely on the contribution of false negatives ignoring false positives altogether our proposed model reduces to a standard topic model. Extensive experiments demonstrate the proposed approach is effective and infers more coherent topics than existing related approaches.

## 1 Introduction

Topic models are ubiquitous probabilistic models for text data suitable for corpus exploration and summarisation as well as for predictive tasks (Blei et al., 2003). The inferred topics are deemed to be useful and meaningful for human interpretation. Accordingly, there is a strong need to develop inexpensive quantitative evaluation methods to assess the quality of the inferred topics efficiently and accurately, because human-based evaluations are slow and elaborate.

Mimno et al. (2011) present a useful data-based quantitative evaluation criterion for measuring quality of the topics. The measure relies on pair-wise word co-occurrence statistics computed efficiently over the corpus and agrees well with human-based topical quality evaluations. Wallach et al. (2009) present evaluation methods based on predictive performance (held-out data likelihood). However, Chang et al. (2009) demonstrate with large-scale human-based evaluations that predictive likelihood may not be a useful criterion; models with better predictive ability may infer less semantically meaningful topics. The finding undermines the core modelling assumptions, complicating development of human-interpretable models. Even though many authors (Arora et al., 2012; AlSumait et al., 2009; Griffiths et al., 2004; Minka and Lafferty, 2002; Teh and Jordan, 2010) have proposed particular topic model variants, based on different modelling assumptions, empirically reporting improved topic coherences, all these assumptions may not be interpreted or justified with a robust quantitative evaluation framework.

In this work, we formulate topic modelling as a novel information retrieval task, where the goal is to retrieve recurring word co-occurrence patterns based on the latent topic representation. We quantify the task performance regarding two types of errors, false negatives (referred to as, misses) and false positives, measured via concepts of recall and precision, respectively. We present a novel topic model that allows the user to trade-off between contributions of the two error types efficiently, and show that taking precision also into account significantly improves topic quality. We show that standard topic models emphasise recall, penalising only misses, at the expense of discarding precision altogether not taking into account false positives.

The paper is structured as follows. Section 2 provides sufficient background on topic modelling and shows that standard topic models emphasise the contribution of misses. In Section 3, we first formulate topic modelling as an information retrieval task and present formulations of recall and precision suitable for the task. Then we present a novel model, that is able to balance between recall and precision, accompanied with an efficient inference algorithm. Section 3.3 discusses relevant related work, Section 4 contains experiments and results, and Section 5 concludes the paper.

**Notation:** Consider $M$ documents $\boldsymbol{y}_m$, where $m = 1, \ldots, M$, such that $y_{m,d}$, where $d = 1, \ldots, D$, denotes a frequency of the $d$th term in the vocabulary for the $m$th document. We denote $N_m$ individual words for the $m$th document as $w_{m,n} \in \{1, D\}$, where $n = 1, \ldots, N_m$, and note that $y_{m,d} = \sum_{n=1}^{N_m} \mathbb{I}[w_{m,n} = d]$, where $\mathbb{I}[\cdot]$ denotes the indicator function taking value one if argument is true and zero otherwise.

## 2   Topic models are recall-biased

Standard topic models, prominently Latent Dirichlet Allocation (LDA; Blei et al., 2003), assume multinomial likelihood

$$\mathcal{L}_m = \prod_{d=1}^{D} q_{m,d}^{y_{m,d}},$$

where $\boldsymbol{q}_m \in \Delta^D$ denotes an unknown expectation parameter of the multinomial distribution, satisfying $q_{m,d} \geq 0$ and $\sum_{d=1}^{D} q_{m,d} = 1$. The goal of topic modelling is, based on the corpus, to infer a set of $K$ topics capturing a lower dimensional representation suitable for summarisation and prediction tasks. Topic models assume the expectations $\boldsymbol{q}_m$ decompose as a linear decomposition

$$\boldsymbol{q}_m = \sum_k \boldsymbol{\eta}_k \theta_{m,k},$$

where $\boldsymbol{\eta}_k \in \Delta^D$, for $k = 1, \ldots, K$, correspond to the topics and $\boldsymbol{\theta}_m \in \Delta^K$ to topic proportions.

We define an empirical word occurrence distribution over the vocabulary for the $m$th document

$$p_{m,d} = y_{m,d}/N_m,$$

noting that $\boldsymbol{q}_m$ should be similar to $\boldsymbol{p}_m$, for $m = 1, \ldots, M$. Because the decomposition is unidentifiable, similarities need to be computed between $\boldsymbol{p}_m$ and $\boldsymbol{q}_m$. Naturally, inferring $\boldsymbol{q}_m$ that is closer to $\boldsymbol{p}_m$ leads to more accurate topics.

The mean multinomial log likelihood,

$$1/N_m \log \mathcal{L}_m = 1/N_m \sum_d y_{m,d} \log q_{m,d} = \sum_d p_{m,d} \log q_{m,d},$$

relates to the KL-divergence between empirical and latent word distributions,

$$KL(\boldsymbol{p}_m, \boldsymbol{q}_m) = \sum_d (p_{m,d} \log p_{m,d} - p_{m,d} \log q_{m,d}) = H_m - 1/N_m \log \mathcal{L}_m, \qquad (1)$$

where $H_m$ is the negative entropy of $\boldsymbol{p}_m$. The asymmetric KL-divergence (1) provides a similarity measure between the empirical and latent distributions, that is sensitive to the contribution of *misses*, corresponding to terms for which $\boldsymbol{p}_m$ are large but the corresponding $\boldsymbol{q}_m$ are small, and, thus, closely relates to the concept of *recall*. The model may infer dense and spurious topics, because $\boldsymbol{q}_m$ must be non-zero for all $\boldsymbol{p}_m > 0$, proportionally to the actual counts. Even though, these topics emphasise recall, they may have very low *precision*, containing intruder terms that capture false similarities.

## 3   Information retrieval aspect

We formulate topic modelling as an information retrieval task: based on the retrieval model $\boldsymbol{q}_m$, the goal is to retrieve co-occurring terms. Here, the $\boldsymbol{p}_m$ represent relevances (that is, empirical co-occurrences) and $\boldsymbol{q}_m$ should be similar to the $\boldsymbol{p}_m$, avoiding errors. We characterise two classes of errors, misses as well as false positives: terms for which $\boldsymbol{p}_m$ are large but $\boldsymbol{q}_m$ are small correspond to misses and terms for which $\boldsymbol{q}_m$ are large but $\boldsymbol{p}_m$ are small correspond to false positives. Naturally, concepts of recall and precision may be quantified with the directed KL divergences, because $KL(\boldsymbol{p}_m, \boldsymbol{q}_m)$ emphasises misses and the reversed divergence $KL(\boldsymbol{q}_m, \boldsymbol{p}_m)$ emphasises false positives. Both measures, or divergences in general, are positive and lower bounded by zero with equivalence if and only if the arguments equal. Mean divergences over documents may be used to assess the performance for the corpus.

### 3.1 Connections to precision and recall for binary relevances

It is useful to consider maximum entropy distributions for $\boldsymbol{p}$ and $\boldsymbol{q}$ to further illustrate connections between the $KL$-divergences and standard recall and precision that are suitable for binary relevances (a term is or is not relevant). The maximum entropy distributions, denoted as $\boldsymbol{p}^*$ and $\boldsymbol{q}^*$, take uniform values over the support of the distributions, denoted as $P$ and $Q$, respectively, whereas the remaining values are (arbitrarily close to) zero. In the following, we denote these zero-probabilities with a very small positive number $\epsilon$, $1 \gg \epsilon \approx 0$, noting that $\log \epsilon \ll 0$.

$KL(\boldsymbol{p}^*, \boldsymbol{q}^*)$ consists of the negative entropy of $\boldsymbol{p}^*$, $\sum_i p_i^* \log p_i^* = -\log |P|$, where $|\cdot|$ computes set cardinality, and the cross-divergence, $\sum_i p_i^* \log q_i^*$, which further decomposes into $|P \cap Q|$ true positives with weight $\frac{1}{|P|} \log \frac{1}{|Q|}$, $|(P \cup Q)^c|$ true negatives[1] with weight $\epsilon \log \epsilon$, $|Q| - |P \cap Q|$ false positives with weight $\epsilon \log \frac{1}{|Q|}$ and $|P| - |P \cap Q|$ misses with weight $\frac{1}{|P|} \log \epsilon$. Because $\epsilon \approx 0$, the divergence is dominated by misses, terms that are in $P$ (relevant) but not in $Q$ (retrieved). Thus,

$$KL(\boldsymbol{p}^*, \boldsymbol{q}^*) = C + \frac{|P \cap Q|}{|P|} \log \epsilon, \tag{2}$$

where $C$ contains the remaining expressions, is proportional to standard recall, proportion of relevant terms that are retrieved.

On the other hand, $KL(\boldsymbol{q}^*, \boldsymbol{p}^*)$ consists of the negative entropy of $\boldsymbol{q}^*$, $-\log |Q|$, and the cross-divergence, $\sum_i q_i^* \log p_i^*$, which decomposes into $|P \cap Q|$ true positives with weight $\frac{1}{|Q|} \log \frac{1}{|P|}$, $|(P \cup Q)^c|$ true negatives with weight $\epsilon \log \epsilon$, $|Q| - |P \cap Q|$ false positives with weight $\frac{1}{|Q|} \log \epsilon$ and $|P| - |P \cap Q|$ misses with weight $\epsilon \log \frac{1}{|P|}$. Following similar reasoning as above, the divergence is dominated by false positives, terms that are in $Q$ (retrieved) but not in $P$ (relevant). Thus,

$$KL(\boldsymbol{q}^*, \boldsymbol{p}^*) = C + \frac{|P \cap Q|}{|Q|} \log \epsilon, \tag{3}$$

is proportional to standard precision, proportion of retrieved terms that are relevant.

Because of the connections (2) and (3), we may interpret the directed divergences as generalisations of the concepts of recall and precision for continuously-valued grades of relevances.

### 3.2 Precision-recall balanced topic model

Following the well-known precision-recall trade-off, we present a new model that is able to compromise between the contributions of misses and false positives, both capturing recurring word co-occurrence patterns and avoiding false similarities. We generalise over standard topic models that are only able to account for misses.

Our model is based on the K-divergence (Lin, 1991),

$$K(\boldsymbol{p}_m, \boldsymbol{q}_m) = \sum_d p_{m,d} \log \frac{p_{m,d}}{(1-\lambda) q_{m,d} + \lambda p_{m,d}} \tag{4}$$

where $0 < \lambda < 1$ is a user-defined parameter. In the following, we show that $\lambda$ intuitively trade-offs the balance between recall and precision; Section 4 further establishes experimental evidence supporting this property. We also show that for this divergence inference can be carried out efficiently.

The K-divergence equals zero if and only if $\boldsymbol{p}_m = \boldsymbol{q}_m$ and is both lower as well as upper bounded,

$$0 \leq K(\boldsymbol{p}_m, \boldsymbol{q}_m) \leq -\log(\lambda).$$

The K-divergence is always well-defined for all values for $\boldsymbol{q}_m \in \Delta$; this is especially relevant at the boundaries of $\Delta$. Consequently, the K-divergence (4) is not as sensitive to misses as $KL(\boldsymbol{p}_m, \boldsymbol{q}_m)$, which approaches infinity close to the boundaries imposing infinite penalty for misses, essentially, imposing a barrier function.

We note that for the maximum entropy distributions, as considered in Section 3.1, the K-divergence becomes, for $\epsilon \to 0$,

$$K(\boldsymbol{p}^*, \boldsymbol{q}^*) = -\frac{|P \cap Q|}{|P|} \log \left( 1 + \widehat{\lambda} \frac{|P \cap Q|}{|Q|} \left( \frac{|P \cap Q|}{|P|} \right)^{-1} \right) - \log(\lambda),$$

where $\widehat{\lambda} = \frac{1-\lambda}{\lambda}$. Applying the logarithmic inequality, $\frac{x}{x+1} < \log(1+x) < x$; $x > -1 \wedge x \neq 0$, we further notice, that the first expression on the right hand side of the divergence is bounded between weighted harmonic mean of precision and recall and weighted precision,

$$\frac{1-\lambda}{\lambda\left(\frac{|P \cap Q|}{|Q|}\right)^{-1} + (1-\lambda)\left(\frac{|P \cap Q|}{|P|}\right)^{-1}} < \frac{|P \cap Q|}{|P|}\log\left(1 + \widehat{\lambda}\frac{|P \cap Q|}{|Q|}\left(\frac{|P \cap Q|}{|P|}\right)^{-1}\right) < \widehat{\lambda}\frac{|P \cap Q|}{|Q|}.$$

For $\lambda$ close to zero, the divergence emphasises recall, whereas for increasing $\lambda$ it takes precision also into account.

We complement the topic model with a mixture of the $\boldsymbol{q}_m$ and document-specific distribution $\boldsymbol{b}_m \in \Delta^D$. For $\boldsymbol{b}_m = \boldsymbol{p}_m$, the corresponding likelihood for the $\boldsymbol{y}_m$ is

$$\mathcal{L}_m^K = \prod_d \left((1-\lambda)q_{m,d} + \lambda b_{m,d}\right)^{y_{m,d}}$$

and we note that the likelihood is connected to the K-divergence,

$$^1/_{N_m}\log\mathcal{L}_m^K = H_m - K(\boldsymbol{p}_m, \boldsymbol{q}_m).$$

In order to retain the properties of the K-divergence suitable for the information retrieval setting considered, we assume $\boldsymbol{b}_m = \boldsymbol{p}_m$, estimating the $\boldsymbol{b}_m$ based on the observed counts. We emphasise that even though we may not generate data from the prior distribution we may use the predictive and posterior distributions as usual. There is little or no need in practice to generate data from the *prior distribution* and all inferences condition on the observed data.

To carry out inference, we apply an MCMC framework in an empirical Bayesian setting, following Casella (2001), employing the empirical distributions as well as introducing prior distributions for the topics as well as topic proportions. We prefer using MCMC over approximate VB or EP approaches that fail to address the true posterior distribution.

We introduce i) word-specific binary assignment variables

$$x_{m,n} \sim \text{Bernoulli}(\lambda),$$

for $m = 1, \ldots, M$ and $n = 1, \ldots, N_m$, to indicate whether $w_{m,n}$ is explained by the $\boldsymbol{q}_m$ or $\boldsymbol{p}_m$ and ii) categorical topic assignment variables $c_{m,n} \in \{1, \ldots, K\}$ for words that are generated based on the $\boldsymbol{q}_m$, respectively. When $x_{m,n} = 0$ with probability $1 - \lambda$, the word is explained by the $\boldsymbol{q}_m$ as in standard topic models. Given the word assignment

$$c_{m,n} \sim \text{Categorical}(\boldsymbol{\theta}_m)$$

the word is generated from the $c_{m,n}$th topic,

$$w_{m,n} \sim \text{Categorical}(\boldsymbol{\eta}_{c_{m,n}}).$$

To complete the model description, we assume

$$\boldsymbol{\eta}_k \sim \text{Dirichlet}(\gamma\mathbf{1}), \quad \boldsymbol{\theta}_m \sim \text{Dirichlet}(\boldsymbol{\alpha}),$$

where $\gamma$ and $\alpha_k$, for $k = 1, \ldots, K$, denote parameters of the Dirichlet distributions.

We present a collapsed Gibbs sampling algorithm building on Griffiths and Steyvers (2004) to carry out posterior computations efficiently. We jointly sample the two types of assignment variables. The probability that $w_{m,n} = d$ is assigned to the $k$th topic is

$$p(c_{m,n} = k, x_{m,n} = 0) \propto \frac{N_{k,m}^{-(w_{m,n})} + \alpha_k}{\sum_{k'} N_{k',m}^{-(w_{m,n})} + \sum_{k'} \alpha_{k'}} \times \frac{G_{k,d}^{-(w_{m,n})} + \gamma}{\sum_{d'} G_{k,d'}^{-(w_{m,n})} + \gamma D}$$

and the probability that the term is explained by the empirical distribution is

$$p(x_{m,n} = 1) \propto \frac{\lambda}{1-\lambda}p_{m,d}.$$

Here the upper index $(\cdot)^{-(w_{m,n})}$ denotes discarding contribution of the current word from topic-document and topic-term count matrices denoted by $N_{k,m}$ and $G_{k,d}$, respectively. Each sampling step updates all the assignments. The algorithm has little additional computational load compared to

collapsed Gibbs sampling for LDA, obtained when setting $\lambda = 0$, because the empirical distributions may be cached.

**Spatio-temporal extension:** In addition to text data, we also demonstrate the model on crime data. Here, the terms correspond to crime occurrences within an area and documents collect occurrences in non-overlapping time windows. Accordingly, to impose smoothness, we modify only the priors for $\boldsymbol{\theta}_m$ and $\boldsymbol{\eta}_k$; the topics may be interpreted as *crime maps*, more mass is assigned to areas with higher crime rates. We introduce $\boldsymbol{\beta}_k \sim \text{Normal}(\mathbf{0}, \mathbf{Q}^{-1})$, for $k = 1, \ldots, K$, and use $\boldsymbol{\eta}_k \propto \exp(\boldsymbol{\beta}_k)^2$. The elements of $\mathbf{Q}$ for off-diagonal elements take value $-\delta$ for two neighbouring areas, otherwise zero, and the diagonal contains the total number of neighbours for each area plus an additive constant $\kappa > 0$ multiplied by $\delta$. We use $\theta_{m,k} \propto \exp(\alpha_{m,k})$, where $\alpha_{m,k} \sim \text{Normal}(\alpha_{m-1,k}, \tau^{-1})$, for $m > 1$, and $\alpha_{1,k} \sim \text{Normal}(0, 10)$. We fix $\kappa$ to a small value ($10^{-2}$) and infer $\alpha_{m,k}$ and $\beta_{k,d}$ using slice sampling, and employ Gibbs for $\delta$ and MH for $\tau$ with $\text{Gamma}(1, 10^{-3})$ priors.

## 3.3 Related work

Chemudugunta et al. (2006) present a related topic model that is able to infer, in addition to topics that are shared by all documents, document-specific distributions that explain document-specific words. Following our model notation, the model introduces $\lambda_m$ for each document and infers $\boldsymbol{b}_m$ based on the data, employing symmetric and weakly informative Beta and Dirichlet priors for the $\lambda_m$ and $\boldsymbol{b}_m$, respectively. For this model, almost surely $\boldsymbol{b}_m \neq \boldsymbol{p}_m$, meaning that the model has no connection to the K-divergence and, importantly, to the information retrieval setting and is unable to balance between precision and recall, as considered in this work. In other words, $\boldsymbol{b}_m$ biases the latent representation $\boldsymbol{q}_m$. Interestingly, we show that our model may be interpreted as a limiting case when adopting strongly informative and asymmetric priors, as follows. Assume

$$\lambda_m \sim \text{Beta}\left((1 - \lambda)\,v, \lambda v\right) \text{ and } \boldsymbol{b}_m \sim \text{Dir}(\boldsymbol{p}_m v + \epsilon \mathbf{1}),$$

where $v$ denotes strength of the prior and $\epsilon \approx 0$. When $v \to \infty$, the priors reduce to point distributions and the model becomes equivalent to our model. Both computationally and conceptually, our model is simpler; in practice, tuning the prior strengths is not straightforward. This tuning can be expensive and is further data-set dependent.

Stochastic Neighbour Embedding (SNE; Hinton and Roweis, 2002) is a statistical model for normalised similarity data suitable for nonlinear dimensionality reduction. The model applies KL-divergences between the observed similarities (distributions) and latent distributions as likelihoods. Peltonen and Kaski (2011) propose a variant of SNE, that applies K-divergence instead, although the authors do not cite the original work of Lin (1991), and show that the model provides improved visualisation performance compared to the original SNE.

# 4 Results

We compare our model against LDA and, as discussed in Section 3.3, to the closely related model by Chemudugunta et al. (2006), referred to as SW model. For all the models, based on text data, we employ collapsed Gibbs sampling for inference.

We quantitatively evaluate topic semantic coherences (Mimno et al., 2011) and entropies, directed KL-divergences, corresponding to concepts of precision and recall, standard recall and precision for binarised relevances as well as (metric) variational ($\ell_1$) distances and adjusted rand index (ARI) for document clustering (when category information is available) for various data collections and for a wide range of different values for $\lambda$.

We compute the divergences and distances for held-out (test) data not used for inferring the topics. We sample $1/5$ of the documents for each data collection to create a test set containing $\widehat{M}$ documents. We estimate the latent test distribution as

$$\widehat{\boldsymbol{q}}_m = \frac{1}{S} \sum_s \sum_k \boldsymbol{\eta}_k^{(s)} \widehat{\theta}_{m,k}^{(s)},$$

averaging over $S$ posterior samples[3]. To ease presentation of results, we denote mean divergences for the held-out data as

$$\text{(mean) recall} = -\frac{1}{\widehat{M}} \sum_m KL(\widehat{\boldsymbol{p}}_m, \widehat{\boldsymbol{q}}_m) \text{ and (mean) precision} = -\frac{1}{\widehat{M}} \sum_m KL(\widehat{\boldsymbol{q}}_m, \widehat{\boldsymbol{p}}_m),$$

where $\widehat{\boldsymbol{p}}_m$ denotes the test empirical distributions. We also evaluate (mean) standard recall and precision; here $P_m$ contains all the terms that occur at least once for the $m$th test document and $Q_m$ contains the top-J retrieved terms based on $\widehat{\boldsymbol{q}}_m$, correspondingly. We note that for both measures higher values indicate better performance. When computing the test divergence $KL(\widehat{\boldsymbol{q}}_m, \widehat{\boldsymbol{p}}_m)$, we smooth the $\widehat{\boldsymbol{p}}_m$ by adding a very small constant to the counts before normalisation in order to prevent numerical problems; the cost of false positives should be large but finite. We also compute (mean) $\ell_1$ distances as

$$\frac{1}{\widehat{M}} \sum_m \sum_d |\widehat{q}_{m,d} - \widehat{p}_{m,d}|.$$

Computation of the topic coherences requires specifying a threshold for sorting $T$ most probable terms for each topic in decreasing order. The measure penalises for intruder and random terms corresponding to false similarities. We show results for $T = \{5, 10, 15, 20\}$. For the entropies, we note that topics with low entropy focus the probability mass on few terms, indicating sparsity; a highly desired property for improving interpretability. We average values for coherences and entropies over the topics. When category information is available, we cluster documents according to the most active topic for each document based on $\boldsymbol{\theta}_m^{(s)}$ and compute adjusted Rand index (ARI) to measure similarity between the inferred clusterings and available category information. We do not assume the number of clusters to be known; the number of potential clusters is constrained by the number of topics.

We show the model performance for three subsets of publicly available data collections, NYTIMES[4], movie reviews[5] and 20newsgroup[6], as well as for textual product descriptions combined with categorical information that we employ for further evaluations. Category information is also available for 20newsgroup. Table 1 shows relevant statistics for each collection.

Table 1: Data statistics.

| Data set | $M$ | $D$ | $\sum_m N_m$ |
|---|---|---|---|
| NYTIMES | 6800 | 19908 | $2.00 \times 10^6$ |
| PRODUCTS | 7743 | 14237 | $1.29 \times 10^6$ |
| MOVIES | 4997 | 25884 | $0.80 \times 10^6$ |
| 20NEWSGROUP | 18307 | 28794 | $2.03 \times 10^6$ |

We initialise the assignments randomly and set $\alpha_k = 0.1$ and $\gamma = 0.01$, corresponding to weakly informative priors, and use $5 \times 10^3$ sampling steps as burnin. After the burnin we collect posterior averages for $S = 200$ samples. We find the number of steps for burnin sufficient for convergence by monitoring log likelihood. We infer the models for $K = 200$ topics and for 21 equi-spaced values between $(0, 0.2)$ for $\lambda$, noting that, $\lambda = 0$, corresponds to the standard topic model (LDA).

Table 2 collects results for $\lambda = 0.1$ for our model, LDA and SW. Unsurprisingly, recall is always best for LDA ($\lambda = 0$). For our model recall decreases, naturally, because the model takes also precision into account; precision is best for our model. The standard recall (R@J) and precision (P@J) measures for $J = 10$ (in percentages) show that standard precision is always best for our model and similarly for standard recall, except for the 20NG data set. The coherences (coh@T) are consistently best for our model, except for the PROD data set for $T = 20$, showing that models that focus solely on recall do not obtain high coherences. This observation is in agreement with Chang et al. (2009), who find that models with better predictive performance (i.e., mean recall) may infer less semantically meaningful topics. Our model also attains smaller (better) mean $\ell_1$ distances, which evaluate metric distance, between $\widehat{\boldsymbol{p}}$ and $\widehat{\boldsymbol{q}}$. Further, the inferred topics of our model are more sparse, measured via topic entropies (ent). In addition, our model attains best ARI

Table 2: Quantitative results for our model for $\lambda = 0.1$, LDA and SW for various data sets. Bolding indicates best results that are statistically significant ($p < 0.01$). For the recall and precision measures and distances we use the paired one-sided Wilcoxon test over the test documents, and for the coherences, entropies and ARI the unpaired one-sided Wilcoxon test over the $S$ samples.

| NYT | recall | precision | R@J[%] | P@J[%] | coh@5 | coh@10 | coh@15 | coh@20 | $\ell_1$ | ent | ARI |
|---|---|---|---|---|---|---|---|---|---|---|---|
| Our | -3.61 | **-72.7** | **5.97** | **63.6** | **-16** | **-77.6** | **-190** | **-358** | **1.63** | **4.78** | - |
| LDA | **-2.97** | -78.8 | 5.22 | 55.6 | -23.1 | -114 | -275 | -511 | 1.74 | 5.85 | - |
| SW | -3.14 | -79.3 | 5.28 | 56.6 | -16.4 | -80.8 | -200 | -378 | 1.75 | 5.82 | - |
| **20NG** | recall | precision | R@J[%] | P@J[%] | coh@5 | coh@10 | coh@15 | coh@20 | $\ell_1$ | ent | ARI |
| Our | -5.54 | **-83.7** | 8.92 | **32.3** | **-18.8** | **-99.2** | **-257** | **-521** | **1.84** | **4.49** | **0.209** |
| LDA | **-4.22** | -86.4 | **9.85** | 30.2 | -21 | -116 | -306 | -606 | 1.9 | 5.81 | 0.15 |
| SW | -4.35 | -86.3 | 9.76 | 30.4 | -20.4 | -106 | -277 | -538 | 1.9 | 5.74 | 0.166 |
| **PROD** | recall | precision | R@J[%] | P@J[%] | coh@5 | coh@10 | coh@15 | coh@20 | $\ell_1$ | ent | ARI |
| Our | -3.62 | **-70.5** | **11.7** | **69.1** | **-16.9** | **-87.1** | **-226** | -462 | **1.56** | **3.6** | **0.15** |
| LDA | **-2.81** | -77.8 | 9.44 | 55.8 | -23.8 | -121 | -304 | -584 | 1.7 | 5 | 0.127 |
| SW | -2.98 | -77.8 | 9.94 | 58.7 | -17.9 | -90.2 | -230 | **-447** | 1.7 | 4.91 | 0.133 |
| **MOV** | recall | precision | R@J[%] | P@J[%] | coh@5 | coh@10 | coh@15 | coh@20 | $\ell_1$ | ent | ARI |
| Our | -4.41 | **-73.7** | **8.53** | **57.2** | **-16.1** | **-88.6** | **-237** | **-493** | **1.67** | **4.89** | - |
| LDA | **-3.61** | -82.6 | 7.91 | 51.5 | -25.3 | -143 | -390 | -787 | 1.82 | 5.87 | - |
| SW | -3.66 | -82.1 | 8.17 | 52.8 | -17.9 | -95.5 | -255 | -522 | 1.81 | 5.73 | - |

values, showing that topics inferred by our model are in closer agreement with the external category information, providing further quantitative evidence of better performance for our model. We note that the conclusions based on Table 2 are similar for $\lambda \in (0.07, 0.11)$, showing that obtaining good results is not sensitive for particular $\lambda$. We also experimented with a variant of the SW model that additionally includes a shared background distribution (referred to as, SWB). The results for SWB are marginally worse or similar to SW, suggesting that including a common background distribution is not effective for improving performance.

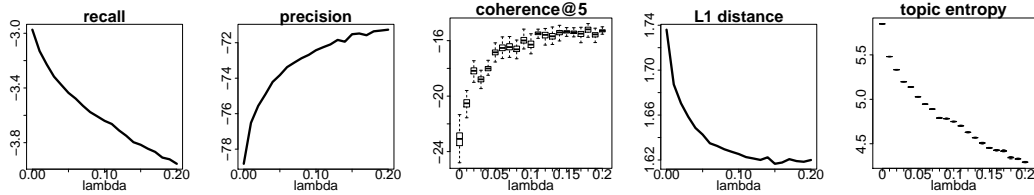

Figure 1: Various performance measures for different values for $\lambda$ for the NYT data set. We note that LDA corresponds to $\lambda = 0$. The curves are similar for the other data sets.

Figure 1 shows results for a wider range of $\lambda$ for the NYT data set. For the divergences and distances we plot mean values versus $\lambda$ and for entropies and coherences we use boxplots, respectively. The performance curves, as shown in Figure 1, are smooth for a wide range of values for $\lambda$, demonstrating stable computations, and we see that recall is always best for LDA ($\lambda = 0$), decreasing for increasing values for $\lambda$, and precision increases for increasing values for $\lambda$. Even small deviations from $\lambda = 0$ are sufficient to shift the focus from recall to a compromise between recall and precision. For increasing $\lambda$ the model also attains smaller (better) mean $\ell_1$ distances. The saturating distance curves also show an effective range for $\lambda$ values; for $\lambda \gg 0.2$ (not shown), the computations eventually become more unstable, because more and more terms are assigned to the empirical distributions and the topics become too sparse complicating posterior inference. In particular, the entropies show how the topics become (on average) more sparse for increasing $\lambda$. The coherences are best for our model for intermediate values for $\lambda$. The coherence curves follow similar trend for other values for the threshold $2 \leq T \leq 20$. However, the measure is sensitive to topic sparsity; if the support of the topic distribution is smaller than $T$ the measure becomes more noisy and less meaningful. We verified that the supports of the topics are larger than $T = 20$ for the data collections for $\lambda \leq 0.2$.

To summarise, we observe a general trend showing that recall is negatively correlated with the other performance measures; as recall decreases, the other measures improve. On the other hand, we observe that precision is positively correlated with the other performance measures, excluding recall. Higher precision implies that the model infers i) latent distributions that have smaller $\ell_1$ distance to test empirical distributions, and ii) topics that are more sparse and more semantically meaningful.

We emphasise that increased sparsity alone does not indicate improved precision; sparse topic models that aim to infer sparse topics focus solely on recall. For instance, Wang and Blei (2009) report improved predictive performance (that is, recall) for sparse models.

Tuning of the LDA hyperparameters is ineffective to trade-off recall and precision. Intuitively, prior tuning is unable to overcome problems of the likelihood function; here the sensitivity of the $KL(\mathbf{p}, \mathbf{q})$ to misses. We fix this issue by modifying directly the likelihood function. We empirically varied both $\alpha$ and $\gamma$ in the range $\{10^{-3}, 10^{-2}, 0.1, 1\}$ for LDA for all data sets and found that $\gamma \leq 0.01$ (inducing sparsity) is preferred for topics (larger values produced useless results despite different $\alpha$). Although, too small $\gamma$ may increase computational complexity and the risk of getting stuck in a locally optimal mode. For a sparse topic prior, increasing $\alpha$ (i) decreases topic entropies (inferring sparser topics), (ii) coherences improve marginally or remain the same and (iii) both recall and precision decrease. Recall and precision curves for different $\gamma$ as a function of $\alpha$ have similar form peaking at $\alpha = \{0.01, 0.1\}$ and $\gamma = \{10^{-3}, 10^{-2}\}$, verifying that the adopted setting for LDA is competitive. Despite the tuning, the precisions and coherences are worse than for our model. We also repeated the experiment for an asymmetric topic prior, that is proportional to overall term occurrences, matching the prior strength to equal the strength of the symmetric variant for varying $\gamma$. The results for the asymmetric prior are very similar to the symmetric prior showing that the asymmetric prior is ineffective to boost precision.

Table 3: Illustration of top topics with top words inferred based on the NYT data.

**Our model**

| |
|---|
| T1 point game team shot half minutes play lead season left rebound games guard coach *laker win quarter night played ball |
| T2 game team playoff season titan games *nfl *jacksonville *miami dolphin play quarterback win *tennessee jaguar *super-bowl *dan-marino played yard won |
| T3 team player game games season play coach played basketball sport fan win playing championship winning guy won record league football |
| T4 *al-gore *bill-bradley *bradley campaign *iowa president democratic *new-hampshire health vice care voter debate caucuses support presidential candidates poll vote administration |
| T5 tablespoon cup minutes add oil pepper large garlic medium serve onion sauce serving bowl fresh pound chopped taste butter chicken |

**LDA**

| |
|---|
| T1 guy right look thought hard talk tell getting put feel bad remember told trying happen kind give real ago sure |
| T2 asked question statement called told saying public interview conference meeting comment reporter added issue took decision member matter plan clear |
| T3 win won record winning victory lost beat past early loss road finished final season home losing start lead close need |
| T4 need feel help problem kind find try getting job able success important step experience level look right start hard hope |
| T5 company companies business industry customer market part high product technology executive firm president competition executives line competitor big chief system |

**SW**

| |
|---|
| T1 company companies business industry million firm customer executive largest executives market billion part analyst chief businesses employees services sales president |
| T2 team season playoff game *nfl games quarterback coach *super-bowl football player *jacksonville titan *miami dolphin *tennessee played play record *ram |
| T3 win lead lost won final beat victory point season record loss early home put winning start gave right consecutive losing |
| T4 guy real right big put look pretty talk course happen tell getting bad mean kid talking wrong hear question head |
| T5 election presidential candidates campaign voter democratic candidate republican vote political *republican primary president race party democrat *party support poll win |

Table 4: Illustration of top terms explained by the empirical distributions (or document-specific distributions) for the NYT data.

| Our model | SW model |
|---|---|
| million percent home plan team right system company | *mccain percent *governor-bush *john-mccain |
| problem part need game official point early money | *bill-bradley *george-bush *bradley women *bush drug |
| american president run play business public record talk | *clinton *al-gore *internet fund *bleated-nato abortion |
| high head set government told place night show big | union *ram *party test *black children card |
| country season decision control deal half return found | *harvard-pilgrim gun *steve-forbes bill *army *gore game |
| look line left find help called family group newspaper | cancer *cowboy *buc firm companies *republican *russia |

Table 3 shows top-5 topics, based on one posterior sample, sorted according to decreasing topic size, another useful measure for topic quality (Mimno et al., 2011), for the NYT data collection, for our model for $\lambda = 0.1$, LDA and SW model. Named entities are referred to as using *-symbol for the terms. The topics for our model are semantically meaningful, capturing certain intuitive themes, as desired. On the other hand, the LDA topics capture frequently occurring words but the topics are not as meaningful and do not correspond to any evident themes. Inspection of such poor quality topics, thought to be the most representative, undermine users' confidence in trusting the inferred model. The SW model falls between our model and LDA, retaining topics similar to LDA that are not meaningful.

Table 4 shows top words assigned to the empirical distributions (or document-specific distributions) $\boldsymbol{b}_m$ for our and SW models. For our model these terms correspond to frequently occurring terms

over the collection that pollute the latent representation of LDA. Our model is able to explain these terms via the empirical distributions leading to more meaningful and more sparse topics, also inferring more accurate latent representations, as verified in Figure 1. The terms for the SW model are document-specific capturing names of persons or places; most of the terms correspond to the named entities. Thus the model still needs to explain the frequently occurring terms over the whole collection, similarly to LDA, inferring poor quality topics that are dense as verified by the large topic entropies. Also, the bias introduced by the $b_m$ leads to inaccurate latent representations as measured in terms of $\ell_1$ distances and the divergences.

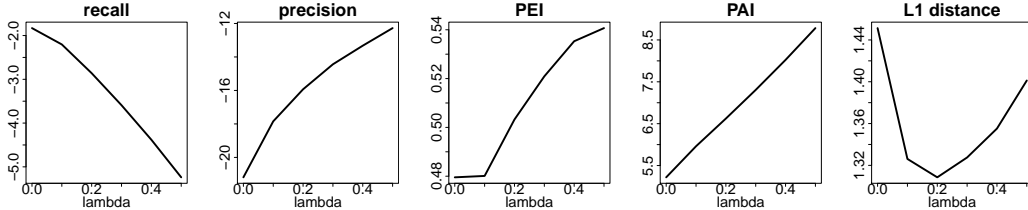

Figure 2: Various performance measures for different values for $\lambda$ for *crime prediction*.

**Spatio-temporal extension:** We use publicly available crime data for London[7] for crime prediction. We discretise the data both in space and time, resulting in $M = 71$ months (finest resolution available), $D = 3.06 \times 10^4$ and $\sum_m N_m = 2.68 \times 10^5$. For the spatial discretisation and computing $\mathbf{Q}$, we use *R-INLA*[8]. Our task is to predict 'hot spots', a collection of mesh points where crime occurs. Following Flaxman et al. (2018), we use predictive efficiency and accuracy indices (PEI and PAI, respectively) for evaluation (the higher, the better). PAI penalises the values by the predicted area size, giving large values for crime hotspots using the smallest area. PEI computes a ratio between the number of crimes occurred in the predicted hotspots and the maximum number of crimes that could have occurred in same area size. In general, PAI and PEI may be interpreted as generalisations of precision and recall, correspondingly, for spatial crime hotspot prediction. We compute the number of hot spots by simulation based on $\widehat{q}_m$, taking mean of non-zero areas (that is, support of the distribution, $|\widehat{Q}_m|$) and set top-$|\widehat{Q}_m|$ areas as hot spots. Again, we remove $1/5$ as test data and estimate $\widehat{q}_m$ by simulation from the posterior. For $K = 5$, Figure 2 shows: i) the recall-precision trade-off, ii) better PEI and PAI for increasing values of $\lambda$ and iii) smaller $\ell_1$ distances for intermediate $\lambda$. For $0.3 \leq \lambda \leq 0.5$, the results are statistically meaningful compared to the LDA variant ($\lambda = 0$)[9]. The conclusions are similar for $K = \{4, 5, \ldots, 10\}$ and the performance does not improve significantly for $K > 5$.

## 5  Discussion

In this work, we present new insights into topic modelling from an information retrieval perspective and propose a novel statistical topic model combined with an efficient inference algorithm that allows the user to balance between contributions of precision and recall, inferring more coherent and meaningful topics. Based on extensive experiments for various data collections and settings, the results demonstrate the proposed approach is effective and useful.

**Acknowledgements**

The authors were supported by the EPSRC grant EP/P020720/1, Inference COmputation and Numerics for Insights into Cities (ICONIC), `https://iconicmath.org/`.

## Footnotes

[1] Upper index $(\cdot)^c$ stands for set complement.

[2] For identifiability, we fix $\beta_{k,1} \approx 0$ by setting the corresponding variance to an arbitrarily small value.

[3]For each sample $\boldsymbol{\eta}_k^{(s)}$, $k = 1, \dots, K$, we sample $\widehat{\boldsymbol{\theta}}_m^{(s)}$.

[4]https://archive.ics.uci.edu/ml/datasets/Bag+of+Words

[5]http://www.cs.cornell.edu/people/pabo/movie-review-data/

[6]http://qwone.com/~jason/20Newsgroups/

[7] `https://data.police.uk/data/`; we collect crimes in *public order*-category.

[8] `http://www.r-inla.org`; we use $2D$ mesh function with 100m cut-off distance between any two mesh points. Areas are parameterised by the mesh points, following the idea of Voronoi tesselation.

[9] Paired one-sided Wilcoxon; $p < 5 \times 10{-4}$.

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
