[Reviews · NeurIPS 2019]

Reviewer 1



Originality * This paper's main contribution of recall-precision balanced topic model is quite original, as no other topic model (AFAIK) tries to balance recall and precision, even though those are widely used and sensible metrics. * The model itself, deriving from KL divergence between the predicted and empirical distributions, and its relationship with precision and recall is simple and elegant. * This paper made me think about sparse topic models, and I am glad this is mentioned in the paper. However, I don't think the authors do enough; just saying that the sparse topic models are evaluated only from the perspective of maximizing recall does not automatically mean that they would do poorly on the precision dimension. I would have liked to see an empirical comparison with a sparse topic model, especially given that there are more advanced sparse models, such as Zhang, et al WWW2013. Quality * The experiments are done well, comparing the three models using a variety of metrics including recall/precision (KL based and conventional), topic coherence, adjusted rand index on classification, and topic entropy. Some of the non-conventional metrics are explained well. * I do have one question about classification results on the datasets that have class labels. Why do you not report precision/recall/F1 scores for classification? Clarity * This paper is quite clear for the most part, though I do not fully understand the short section about the crime rate. Since this part is, application-wise, quite different from text modeling, more friendly description, rather than pointing to a reference paper, would be helpful (e.g., what PEI and PAI mean). Significance * I do have one concern about this paper. As much as topic modeling is an important subfield within ML, I keep wondering how significant this paper will be. Will it be cited and used for further research in latent semantics of text? Given many of the text modeling tasks are done with neural network based models (often in combination with latent variables), it would be helpful for the authors to explain and emphasize the significance of this research. ** Post-author response ** I appreciate the authors addressing the questions and concerns in my review.

Reviewer 2



The authors motivate the problem quite effectively and go into detail on the recall bias (that is, propensity of inference methods to penalize topic distributions significantly for missing document terms) of standard topic models. This is a useful observation that can motivate researchers to address this inherent bias. The authors propose a new topic model to balance precision and recall. This is done by introducing a Bernoulli distribution that controls word token generation through topics or through document specific word distribution. The authors note that this is similar to the SW model by Chemudugunta et al. The differences are that the SW model uses a weak symmetric prior for the Bernoulli distribution (\lambda) and uses a biased estimate for the document-specific word distribution. Experimental results measuring precision, recall, coherence, etc. demonstrate that the proposed model is significantly better on all metrics except recall (as one would expect). This is a significant result. I have some questions that I would like the authors to respond/address: In my opinion, the differences between the proposed model and the SW model are not significant. For example, it is straightforward to convert a weak symmetric prior to a strong asymmetric beta prior in the current setting. Maybe the novelty is the way the document-specific word distributions are generated and the theoretical connection of your proposed model to the K-divergence. (ii) Secondly, based on the reasoning in lines 268-273, it looks like one other disadvantage of the SW model is its inability to explain generic words in the corpus. However, the same paper also introduces SWB model to address this issue. It would be useful to compare your model to the SWB version. (iii) It would be useful to see if the results in Table 2 are sensitive to \lambda. My understanding is that all of them use \lambda = 0.1. The paper is quite well-written and the theoretical motivation for proposing the model is compelling.

Reviewer 3



Thanks to the authors for the helpful response and the additional experiments. Original Review: This paper makes an argument that standard LDA is biased in favor of recall, and it presents a method that can remove the bias. In experiments, the new method performs well on a precision metric and on topic coherence. This paper seems to be making an interesting insight. However, I had a hard time understanding the arguments, and I think the paper’s analysis and experiments do not sufficiently evaluate how much the recall focus of LDA depends on specific choices for hyperparameters or symmetric priors. The derivation around Equation 1 is true for any model that is trained to maximize likelihood, so when the paper declares there that Equation 1 is “sensitive to misses” it is hard to understand why. It is not until later that the paper aims to explain, but the explanation comes only in terms of a uniform model and the argument is made informally. It would help me if the paper stated its results more formally, as a conjecture or theorem with proof, so that I can better know what is being claimed, and to what extent the result relies on assumptions like uniformity. Also, the paper would be stronger if its mathematical illustrations used more realistic settings (e.g. Zipfian distributions for word frequency). The algorithm ends up being a fairly simple variant on LDA that mixes in a document-specific distribution (estimated from the ground truth document counts) along with the standard topic model. As the paper says, this means that the model is not a complete generative model of documents, because its likelihood requires knowing the counts of words in each document (b_m) in advance. In the experiments, the paper does not consider altering the LDA hyperparameters. This is a concern because the fact that precision and recall are monotonic in topic entropy (Fig 1) suggests that simply tuning the LDA hyperparameters toward lower-entropy topics might serve to boost precision at the cost of recall. In particular, allowing an asymmetric gamma that reflects corpus-wide word frequency would start to look somewhat similar to this paper’s model, if I am not mistaken, and I think the paper needs to experiment with that approach. Minor: The empirical results state that in their experiment “LDA topics capture frequently occurring words but the topics are not as meaningful and do not correspond to any evident themes”---we know that very often LDA does infer meaningful topics, so investigating and explaining why it failed here would be helpful.

[Author Response · NeurIPS 2019]

We thank the reviewers for their useful feedback. Please find below responses to the comments by each reviewer. Also kindly note that the new results referred to in the following will be added to the final version of the paper.

**R1 and R3**: We empirically varied both $\alpha$ and $\gamma$ in the range $\{10^{-3}, 10^{-2}, 0.1, 1\}$ for LDA for all data sets. We found that $\gamma \leq 0.01$ (inducing sparsity) is preferred for topics (larger values produced useless results despite different $\alpha$). Although too small $\gamma$ increases computational complexity and the risk of getting stuck in a locally optimal mode. For a sparse topic prior, increasing $\alpha$ (i) decreases topic entropies (inferring sparser topics), (ii) coherences improve marginally or remain the same and (iii) **both recall and precision decrease**. Recall and precision curves for different $\gamma$ as a function of $\alpha$ have similar form peaking at $\alpha = \{0.01, 0.1\}$ and $\gamma = \{10^{-3}, 10^{-2}\}$, verifying that the adopted setting for LDA is competitive. Despite the tuning, the precisions and coherences are worse than for our model. As suggested by the reviewer 3, we repeated the experiment for an asymmetric topic prior, matching the prior strength to equal the strength of the symmetric variant for varying $\gamma$. The results for the asymmetric prior are very similar to the symmetric prior showing that the asymmetric prior is ineffective to boost precision. To conclude, tuning of the LDA hyperparameters is ineffective to trade-off recall and precision. Intuitively, prior tuning is unable to overcome problems of the likelihood function; here the sensitivity of the $\text{KL}(\mathbf{p}, \mathbf{q})$ to misses. We fix this issue by modifying directly the likelihood function.

**R1**: We emphasise that we are essentially clustering the documents based on the inferred topics and using ARI to compute similarity between inferred and true clusters. As opposed to classification we do not assume the number of clusters to be known; the number of potential clusters is constrained by the number of topics. We argue the clustering scenario is more interesting than the classification set-up, which additionally induces classification algorithm bias.

Predictive accuracy index (PAI) penalises the values by the predicted area size, giving large values for crime hotspots using the smallest area. Predictive efficiency index (PEI) computes a ratio between the number of crimes occurred in the predicted hotspots and the maximum number of crimes that could have occurred in same area size. In general, PAI and PEI may be interpreted as generalisations of precision and recall, correspondingly, for spatial crime hotspot prediction.

Given the ease of implementation and insensitivity of setting $\lambda$ for obtaining better performance than LDA, which is found in nearly every data scientists' toolbox, we argue our new topic model would serve as a new standard for topic modelling.

**R2**: We found that the results for SWB are marginally worse or similar to SW, suggesting that including a common background term is not effective for improving performance, as suggested by the reviewer. The key difference between SW(B) and our model is specification of the background distributions and $\lambda$. We point out that our model performs better than SW(B) model that has no connection to K-divergence and is not able to trade-off recall and precision.

We agree that it is in theory straightforward to place an informative prior for $\lambda$ instead of fixing a value for it but tuning of the corresponding hyperparameters is less straightforward in practice; in addition to mean also the strength of the prior needs to be specified. This tuning can be expensive and is further data-set dependent.

The conclusions drawn from Table 2 hold exactly for $\lambda \in (0.07, 0.11)$, further showing that obtaining good results is not sensitive for particular $\lambda$. For larger $\lambda$ the coherences for larger thresholds ($T$) may become less meaningful because of increased topic sparsity, noting that coherence computation requires ordering the top-$T$ words. Focusing for smaller thresholds (here for $T = 5$), the conclusions hold for $\lambda \in (0.07, 0.14)$.

**R3**: We adopted fixed symmetric priors for computational reasons, for permitting a fair comparison to SW(B) models and for experimenting the effect of different hyperparameters for LDA (see above).

We use the uniformity assumption to demonstrate the connection between KL-divergences to standard recall and precision, that only work for binary relevances; a term is either relevant or not. KL-divergences are suitable for graded relevances and do not assume uniformity; here the $\log$-function acts as a barrier function giving a large penalty for $p \log(q)$ for non-zero $p$ and small $q$ (corresponding to misses).

Often hand-selected topics, that make sense, are shown in research papers avoiding the issue that top topics according to cardinality may not be meaningful.

[Meta-Review · NeurIPS 2019]

The paper offers a new evaluation criterion for topic models, based on a precision/recall trade off. This criterion is operationalized in a new implementation. Topic modeling is a well-studied field, so finding promising new directions is exciting. The reviewers found the author response compelling and are all in favor of acceptance.